# Unsupervised Cross-Modal Alignment of Speech and Text Embedding Spaces

**Yu-An Chung, Wei-Hung Weng, Schrasing Tong,** and **James Glass**
Computer Science and Artificial Intelligence Laboratory
Massachusetts Institute of Technology
Cambridge, MA 02139, USA
{andyyuan,ckbjimmy,st9,glass}@mit.edu

## Abstract

Recent research has shown that word embedding spaces learned from text corpora of different languages can be aligned without any parallel data supervision. Inspired by the success in unsupervised cross-lingual word embeddings, in this paper we target learning a *cross-modal* alignment between the embedding spaces of speech and text learned from corpora of their respective modalities in an unsupervised fashion. The proposed framework learns the individual speech and text embedding spaces, and attempts to align the two spaces via adversarial training, followed by a refinement procedure. We show how our framework could be used to perform spoken word classification and translation, and the experimental results on these two tasks demonstrate that the performance of our unsupervised alignment approach is comparable to its supervised counterpart. Our framework is especially useful for developing automatic speech recognition (ASR) and speech-to-text translation systems for low- or zero-resource languages, which have little parallel audio-text data for training modern supervised ASR and speech-to-text translation models, but account for the majority of the languages spoken across the world.

## 1 Introduction

Word embeddings—continuous-valued vector representations of words—are almost ubiquitous in recent natural language processing research. Most successful methods for learning word embeddings [1, 2, 3] rely on the distributional hypothesis [4], i.e., words occurring in similar contexts tend to have similar meanings. Exploiting word co-occurrence statistics in a text corpus leads to word vectors that reflect semantic similarities and dissimilarities: similar words are geometrically close in the embedding space, and conversely, dissimilar words are far apart.

Continuous word embedding spaces have been shown to exhibit similar structures across languages [5]. The intuition is that most languages share similar expressive power and are used to describe similar human experiences across cultures; hence, they should share similar statistical properties. Inspired by the notion, several studies have focused on designing algorithms that exploit this similarity to learn a cross-lingual alignment between the embedding spaces of two languages, where the two embedding spaces are trained from independent text corpora [6, 7, 8, 9, 10, 11, 12]. In particular, recent research has shown that such cross-lingual alignments can be learned without relying on any form of bilingual supervision [13, 14, 15], and has been applied to training neural machine translation (NMT) systems in a completely unsupervised fashion [16, 17]. This eliminates the need for a large parallel training corpus to train NMT systems.

Speech, as another form of language, is rarely considered as a source for learning semantics, compared to text. Although there is work that explores the concept of learning vector representations from

speech [18, 19, 20, 21, 22, 23], they are primarily based on acoustic-phonetic similarity, and aim to represent the way a word sounds rather than its meaning.

Recently, the Speech2Vec [24] model was developed to be capable of representing audio segments excised from a speech corpus as fixed dimensional vectors that contain semantic information of the underlying spoken words. The design of Speech2Vec is based on a Recurrent Neural Network (RNN) Encoder-Decoder framework [25, 26], and borrows the methodology of Skip-grams or continuous bag-of-words (CBOW) from Word2Vec [1] for training. Since Speech2Vec and Word2Vec share the same training methodology and speech and text are similar media for communicating, the two embedding spaces learned respectively by Speech2Vec from speech and Word2Vec from text are expected to exhibit similar structure.

Motivated by the recent success in unsupervised cross-lingual alignment [13, 15, 14] and the assumption that the embedding spaces of the two modalities (speech and text) share similar structure, we are interested in learning an unsupervised *cross-modal* alignment between the two spaces. Such an alignment would be useful for developing automatic speech recognition (ASR) and speech-to-text translation systems for low- or zero-resource languages that lack parallel corpora of speech and text for training. In this paper, we propose a framework for unsupervised cross-modal alignment, borrowing the methodology from unsupervised cross-lingual alignment presented in [14]. The framework consists of two steps. First, it uses Speech2Vec [24] and Word2Vec [1] to learn the individual embedding spaces of speech and text. Next, it leverages adversarial training to learn a linear mapping from the speech embedding space to the text embedding space, followed by a refinement procedure.

The paper is organized as follows. Section 2 describes how we obtain the speech embedding space in a completely unsupervised manner using Speech2Vec. Next, we present our unsupervised cross-modal alignment approach in Section 3. In Section 4, we describe the tasks of spoken word classification and translation, which are similar to ASR and speech-to-text translation, respectively, except that now the input are audio segments corresponding to words. We then evaluate the performance of our unsupervised alignment on the two tasks and analyze our results in Section 5. Finally, we conclude and point out some interesting future work possibilities in Section 6. To the best of our knowledge, this is the first work that achieves fully unsupervised spoken word classification and translation.

## 2   Unsupervised Learning of the Speech Embedding Space

Recently, there is an increasing interest in learning the semantics of a language directly, and only from raw speech [24, 27, 28]. Assuming utterances in a speech corpus are already pre-segmented into audio segments corresponding to words using word boundaries obtained by forced alignment, existing approaches aim to represent each audio segment as a fixed dimensional embedding vector, with the hope that the embedding is able to capture the semantic information of the underlying spoken word. However, some supervision leaks into the learning process through the use of forced alignment, rendering the approaches not fully unsupervised.

In this paper, we use Speech2Vec [24], a recently proposed deep neural network architecture that has been shown capable of capturing the semantics of spoken words from raw speech, for learning the speech embedding space. To eliminate the need of forced alignment, we propose a simple pipeline for training Speech2Vec in a totally unsupervised manner. We briefly review Speech2Vec in Section 2.1, and introduce the unsupervised pipeline in Section 2.2.

### 2.1   Speech2Vec

In text, a Word2Vec [1] model is a shallow, two-layer fully-connected neural network that is trained to reconstruct the contexts of words. There are two methodologies for training Word2Vec: Skip-grams and CBOW. The objective of Skip-grams is for each word $\mathbf{w}^{(n)}$ in a text corpus, the model is trained to maximize the probability of words $\{\mathbf{w}^{(n-k)}, \ldots, \mathbf{w}^{(n-1)}, \mathbf{w}^{(n+1)}, \ldots, \mathbf{w}^{(n+k)}\}$ within a window of size $k$ given $\mathbf{w}^{(n)}$. The objective of CBOW, on the other hand, aims to infer the current word $\mathbf{w}^{(n)}$ from its nearby words $\{\mathbf{w}^{(n-k)}, \ldots, \mathbf{w}^{(n-1)}, \mathbf{w}^{(n+1)}, \ldots, \mathbf{w}^{(n+k)}\}$.

Speech2Vec [24], inspired by Word2Vec, borrows the methodology of Skip-grams or CBOW for training. Unlike text, where words are represented by one-hot vectors as input and output for training Word2Vec, an audio segment is represented by a variable-length sequence of acoustic

features, $\mathbf{x} = (\mathbf{x}_1, \mathbf{x}_2, \ldots, \mathbf{x}_T)$, where $\mathbf{x}_t$ is the acoustic feature such as Mel-Frequency Cepstral Coefficients at time $t$, and $T$ is the length of the sequence. In order to handle variable-length input and output sequences of acoustic features, Speech2Vec replaces the two fully-connected layers in the Word2Vec model with a pair of RNNs, one as an Encoder and the other as a Decoder [25, 26]. When training Speech2Vec with Skip-grams, the Encoder RNN takes the audio segment (corresponding to the current word) as input and encodes it into a fixed dimensional embedding $\mathbf{z}^{(n)}$ that represents the entire input sequence $\mathbf{x}^{(n)}$. Subsequently, the Decoder RNN aims to reconstruct the audio segments $\{\mathbf{x}^{(n-k)}, \ldots, \mathbf{x}^{(n-1)}, \mathbf{x}^{(n+1)}, \ldots, \mathbf{x}^{(n+k)}\}$ (corresponding to nearby words) within a window of size $k$ from $\mathbf{z}^{(n)}$. Similar to the concept of training Word2Vec with Skip-grams, the intuition behind this methodology is that, in order to successfully decode nearby audio segments, the encoded embedding $\mathbf{z}^{(n)}$ should contain sufficient semantic information of the current audio segment $\mathbf{x}^{(n)}$. In contrast to training Speech2Vec with Skip-grams that aims to predict nearby audio segments from $\mathbf{z}^{(n)}$, training Speech2Vec with CBOW sets $\mathbf{x}^{(n)}$ as the target and aims to infer it from nearby audio segments. By using the same training methodology (Skip-grams or CBOW) as Word2Vec, it is reasonable to assume that the embedding space learned by Speech2Vec from speech exhibits similar structure to that learned by Word2Vec from text.

After training the Speech2Vec model, each audio segment is transformed into an embedding vector that contains the semantic information of the underlying word. In a Word2Vec model, the embedding for a particular word is deterministic, which means that every instance of the same word will be represented by one, and only one, embedding vector. In contrast, for audio segments every instance of a spoken word is different (due to speaker, channel, and other contextual differences, etc.), so every instance of the same underlying word is represented by a different (though hopefully similar) embedding vector. Embedding vectors of the same spoken words can be averaged to obtain a single word embedding based on the identity of each audio segment, as is done in [24].

## 2.2 Unsupervised Speech2Vec

Speech2Vec and Word2Vec learn the semantics of words by making use of the co-occurrence information in their respective modalities, and are both intrinsically unsupervised. However, unlike text where the content can be easily segmented into word-like units, speech has a continuous form by nature, making the word boundaries challenging to locate. All utterances in the speech corpus are assumed to be perfectly segmented into audio segments based on the word boundaries obtained by forced alignment with respect to the reference transcriptions [24]. Such an assumption, however, makes the process of learning word embeddings from speech not truly unsupervised.

Unsupervised speech segmentation is a core problem in zero-resource speech processing in the absence of transcriptions, lexicons, or language modeling text. Early work mainly focused on unsupervised term discovery, where the aim is to find word- or phrase-like patterns in a collection of speech [29, 30]. While useful, the discovered patterns are typically isolated segments spread out over the data, leaving much speech as background. This has prompted several studies on *full-coverage* approaches, where the entire speech input is segmented into word-like units [31, 32, 33, 34].

In this paper, we use an off-the-shelf, full-coverage, unsupervised segmentation system for segmenting our data into word-like units. Three representative systems are explored in this paper. The first one, referred to as Bayesian embedded segmental Gaussian mixture model (BES-GMM) [35], is a probabilistic model that represents potential word segments as fixed-dimensional acoustic word embeddings [23], and builds a whole-word acoustic model in this embedding space while jointly doing segmentation. The second one, called embedded segmental K-means model (ES-KMeans) [36], is an approximation to BES-GMM that uses hard clustering and segmentation, rather than full Bayesian inference. The third one is the recurring syllable-unit segmenter called SylSeg [37], a fast and heuristic method that applies unsupervised syllable segmentation and clustering, to predict recurring syllable sequences as words.

After training the Speech2Vec model using the audio segments obtained by an unsupervised segmentation method, each audio segment is then transformed into an embedding that contains the semantic information about the segment. Since we do not know the identity of the embeddings, we use the k-means algorithm to cluster them into $K$ clusters, potentially corresponding to $K$ different word types. We then average all embeddings that belong to the same cluster (potentially the instances of

the same underlying word) to obtain a single embedding. Note that by doing so, it is possible that we group the embeddings corresponding to different words that are semantically similar into one cluster.

# 3 Unsupervised Alignment of Speech and Text Embedding Spaces

Suppose we have speech and text embedding spaces trained on independent speech and text corpora. Our goal is to learn a mapping, without using any form of cross-modal supervision, between them such that the two spaces are aligned.

Let $\mathcal{S} = \{s_1, s_2, \ldots, s_m\} \subseteq \mathbb{R}^{d_1}$ and $\mathcal{T} = \{t_1, t_2, \ldots, t_n\} \subseteq \mathbb{R}^{d_2}$ be two sets of $m$ and $n$ word embeddings of dimensionality $d_1$ and $d_2$ from the speech and text embedding spaces, respectively. Ideally, if we have a known dictionary that specifies which $s_i \in \mathcal{S}$ corresponds to which $t_j \in \mathcal{T}$, we can learn a linear mapping $W$ between the two embedding spaces such that

$$W^* = \underset{W \in \mathbb{R}^{d_2 \times d_1}}{\operatorname{argmin}} \|WX - Y\|^2, \tag{1}$$

where $X$ and $Y$ are two aligned matrices of size $d_1 \times k$ and $d_2 \times k$ formed by $k$ word embeddings selected from $\mathcal{S}$ and $\mathcal{T}$, respectively. At test time, the transformation result of any audio segment $a$ in the speech domain can be defined as $\operatorname{argmax}_{t_j \in \mathcal{T}} \cos(Ws_a, t_j)$. In this paper, we show how to learn this mapping $W$ without using any cross-modal supervision. The proposed framework, inspired by [14], consists of two steps: domain-adversarial training for learning an initial proxy of $W$, followed by a refinement procedure which uses the words that match the best to create a synthetic parallel dictionary for applying Equation 1.

## 3.1 Domain-Adversarial Training

The intuition behind this step is to make the mapped $\mathcal{S}$ and $\mathcal{T}$ indistinguishable. We define a discriminator, whose goal is to discriminate between elements randomly sampled from $W\mathcal{S} = \{Ws_1, Ws_2, \ldots, Ws_m\}$ and $\mathcal{T}$. The mapping $W$, which can be viewed as the generator, is trained to prevent the discriminator from making accurate predictions. This is a two-player game, where the discriminator aims at maximizing its ability to identify the origin of an embedding, and $W$ aims at preventing the discriminator from doing so by making $W\mathcal{S}$ and $\mathcal{T}$ as *similar* as possible. Given the mapping $W$, the discriminator, parameterized by $\theta_D$, is optimized by minimizing the following objective function:

$$\mathcal{L}_D(\theta_D|W) = -\frac{1}{m}\sum_{i=1}^{m}\log P_{\theta_D}(\text{speech} = 1|Ws_i) - \frac{1}{n}\sum_{j=1}^{n}\log P_{\theta_D}(\text{speech} = 0|t_j), \tag{2}$$

where $P_{\theta_D}(\text{speech} = 1|v)$ is the probability that vector $v$ originates from the speech embedding space (as opposed to an embedding from the text embedding space). Given the discriminator, the mapping $W$ aims to fool the discriminator's ability to accurately predict the original domain of the embeddings by minimizing the following objective function:

$$\mathcal{L}_W(W|\theta_D) = -\frac{1}{m}\sum_{i=1}^{m}\log P_{\theta_D}(\text{speech} = 0|Ws_i) - \frac{1}{n}\sum_{j=1}^{n}\log P_{\theta_D}(\text{speech} = 1|t_j) \tag{3}$$

The discriminator $\theta_D$ and the mapping $W$ are optimized iteratively to respectively minimize $\mathcal{L}_D$ and $\mathcal{L}_W$ following the standard training procedure of adversarial networks [38].

## 3.2 Refinement Procedure

The domain-adversarial training step learns a rotation matrix $W$ that aligns the speech and text embedding spaces. To further improve the alignment, we use the $W$ learned in the domain-adversarial training step as an initial proxy and build a synthetic parallel dictionary that specifies which $s_i \in \mathcal{S}$ corresponds to which $t_j \in \mathcal{T}$.

To ensure a high-quality dictionary, we consider the most frequent words from $\mathcal{S}$ and $\mathcal{T}$, since more frequent words are expected to have better quality of embedding vectors, and only retain their mutual nearest neighbors. For deciding mutual nearest neighbors, we use the Cross-Domain Similarity Local Scaling proposed in [14] to mitigate the so-called hubness problem [39] (points tending to be nearest neighbors of many points in high-dimensional spaces). Subsequently, we apply Equation 1 on this generated dictionary to refine $W$.

# 4 Spoken Word Classification and Translation

Conventional hybrid ASR systems [40] and recent end-to-end ASR models [41, 42, 43, 44] rely on a large amount of parallel audio-text data for training. However, most languages spoken across the world lack parallel data, so it is no surprise that only very few languages support ASR. It is the same story for speech-to-text translation [45], which typically pipelines ASR and machine translation, and could be even more challenging to develop as it requires both components to be well trained. Compared to parallel audio-text data, the cost of accumulating independent corpora of speech and text is significantly lower. With our unsupervised cross-modal alignment approach, it becomes feasible to build ASR and speech-to-text translation systems using independent corpora of speech and text only, a setting suitable for low- or zero-resource languages.

Since a cross-modal alignment is learned to link the *word* embedding spaces of speech and text, we perform the tasks of spoken word classification and translation to directly evaluate the effectiveness of the alignment. The two tasks are similar to standard ASR and speech-to-text translation, respectively, except that now the input is an audio segment corresponding to a word.

## 4.1 Spoken Word Classification

The goal of this task is to recognize the underlying spoken word of an input audio segment. Suppose we have two independent corpora of speech and text that belong to the same language. The speech and text embedding spaces, denoted by $\mathcal{S}$ and $\mathcal{T}$, can be obtained by training Speech2Vec and Word2Vec on the respective corpus. The alignment $W$ between $\mathcal{S}$ and $\mathcal{T}$ can be learned in an either supervised or unsupervised way. At test time, given an input audio segment, it is first transformed into an embedding vector $s$ in the speech embedding space $\mathcal{S}$ by Speech2Vec. The vector $s$ is then mapped to the text embedding space as $t_s = Ws \in \mathcal{T}$. In $\mathcal{T}$, the word that has embedding vector $t^* = \operatorname{argmax}_{t \in \mathcal{T}} \cos(t, t_s)$ closest to $t_s$ will be taken as the classification result. The performance is measured by accuracy.

## 4.2 Spoken Word Translation

This task is similar to the one in the text domain that considers the problem of retrieving the translation of given source words, except that the source words are in the form of audio segments. Spoken word translation can be performed in the exact same way as spoken word classification, but the speech and text corpora belong to different languages. At test time, we follow the standard practice of word translation and measure how many times one of the correct translations (in text) of the input audio segment is retrieved, and report precision@ $k$ for $k = 1$ and 5. We use the bilingual dictionaries provided by [14] to obtain the correct translations of a given source word.

# 5 Experiments

In this section, we empirically demonstrate the effectiveness of our unsupervised cross-modal alignment approach on spoken word classification and translation introduced in Section 4.

## 5.1 Datasets

For our experiments, we used English and French LibriSpeech [46, 47], and English and German Spoken Wikipedia Corpora (SWC) [48]. All corpora are read speech, and come with a collection of utterances and the corresponding transcriptions. For convenience, we denote the speech and text data of a corpus in uppercase and lowercase, respectively. For example, $\text{EN}_{\text{swc}}$ and $\text{en}_{\text{swc}}$ represent the speech and text data, respectively, of English SWC. In Table 1, column

Table 1: The detailed statistics of the corpora.

| Corpus | Train | Test | Words | Segments |
|---|---|---|---|---|
| English LibriSpeech | 420 hr | 50 hr | 37K | 468K |
| French LibriSpeech | 200 hr | 30 hr | 26K | 260K |
| English SWC | 355 hr | 40 hr | 25K | 284K |
| German SWC | 346 hr | 40 hr | 31K | 223K |

Train is the size of the speech data used for training the speech embeddings; column Test is the size of the speech data used for testing, where the corresponding number of audio segments (i.e., spoken

word tokens) is specified in column Segments; column Words provides the number of distinct words in that corpus. Train and test sets are split in a way so that there are no overlapping speakers.

## 5.2 Details of Training and Model Architectures

The speech embeddings were trained using Speech2Vec with Skip-grams by setting the window size $k$ to three. The Encoder is a single-layer bidirectional LSTM, and the Decoder is a single-layer unidirectional LSTM. The model was trained by stochastic gradient descent (SGD) with a fixed learning rate of $10^{-3}$. The text embeddings were obtained by training Word2Vec on the transcriptions using the fastText implementation without subword information [3]. The dimension of both speech and text embeddings is 50.[1]

For the adversarial training, the discriminator was a two-layer neural network of size 512 with ReLU as the activation function. Both the discriminator and $W$ were trained by SGD with a fixed learning rate of $10^{-3}$. For the refinement procedure, we used the default setting specified in [14].[2]

## 5.3 Comparing Methods

Table 2: Different configurations for training Speech2Vec to obtain the speech embeddings with decreasing level of supervision. The last column specifies whether the configuration is unsupervised.

| Configuration | Speech2Vec training | | Unsupervised |
|---|---|---|---|
| | How word segments were obtained | How embeddings were grouped together | |
| $A$ & $A^*$ | Forced alignment | Use word identity | ✗ |
| $B$ | Forced alignment | k-means | ✗ |
| $C$ | BES-GMM [35] | k-means | ✓ |
| $D$ | ES-KMeans [36] | k-means | ✓ |
| $E$ | SylSeg [37] | k-means | ✓ |
| $F$ | Equally sized chunks | k-means | ✓ |

**Alignment-Based Approaches**   Given the speech and text embeddings, alignment-based approaches learn the alignment between them in an either supervised or unsupervised way; for an input audio segment, they perform spoken word classification and translation as described in Section 4.

By varying how word segments were obtained before being fed to Speech2Vec and how the embeddings were grouped together, the level of supervision is gradually decreased towards a fully unsupervised configuration. In configuration $A$, the speech training data was segmented into words using forced alignment with respect to the reference transcription, and the embeddings of the same word were grouped together using their word identities. In configuration $B$, the word segments were also obtained by forced alignment, but the embeddings were grouped together by performing k-means clustering. In configurations $C$, $D$, and $E$, the speech training data was segmented into word-like units using different unsupervised segmentation algorithms described in Section 2.2. Configuration $F$ serves as a baseline by naively segmenting the speech training data into equally sized chunks. Unlike configurations $A$ and $B$, configurations $C$, $D$, $E$, and $F$ did not require the reference transcriptions to do forced alignment and the embeddings were grouped together by performing k-means clustering, and are thus unsupervised. Configurations $A$ to $F$ all used our unsupervised alignment approach to align the speech and text embedding spaces.

We also implemented configuration $A^*$, which trained Speech2Vec in the same way as configuration $A$, but learned the alignment using a parallel dictionary as cross-modal data supervision. The different configurations are summarized in Table 2.

**Word Classifier**   We established an upper bound by using the fully-supervised Word Classifier that was trained to map audio segments directly to their corresponding word identities. The Word Classifier was composed of a single-layer bidirectional LSTM with a softmax layer appended at the output of its last time step. This approach is specific to spoken word classification.

**Majority Word Baseline**   For both spoken word classification and translation tasks, we implemented a straightforward baseline dubbed Major-Word, where for classification, it always predicts the most frequent word, and for translation, it always predicts the most commonly paired word. Results of the Major-Word offer us insight into the word distribution of the test set.

## 5.4   Results and Discussion

Table 3: Accuracy on spoken word classification. $EN_{ls} - en_{swc}$ means that the speech and text embeddings were learned from the speech training data of English LibriSpeech and text training data of English SWC, respectively, and the testing audio segments came from English LibriSpeech. The same rule applies to Table 5 and Table 6. For the Word Classifier, $EN_{ls} - en_{swc}$ and $EN_{swc} - en_{ls}$ could not be obtained since it requires parallel audio-text data for training.

| Corpora | $EN_{ls} - en_{ls}$ | $FR_{ls} - fr_{ls}$ | $EN_{swc} - en_{swc}$ | $DE_{swc} - de_{swc}$ | $EN_{ls} - en_{swc}$ | $EN_{swc} - en_{ls}$ |
|---|---|---|---|---|---|---|
| *Nonalignment-based approach* | | | | | | |
| Word Classifier | 89.3 | 83.6 | 86.9 | 80.4 | – | – |
| *Alignment-based approach with cross-modal supervision (parallel dictionary)* | | | | | | |
| $A^*$ | 25.4 | 27.1 | 29.1 | 26.9 | 21.8 | 23.9 |
| *Alignment-based approaches without cross-modal supervision (our approach)* | | | | | | |
| $A$ | 23.7 | 24.9 | 25.3 | 25.8 | 18.3 | 21.6 |
| $B$ | 19.4 | 20.7 | 22.6 | 21.5 | 15.9 | 17.4 |
| $C$ | 10.9 | 12.6 | 14.4 | 13.1 | 6.9 | 8.0 |
| $D$ | 11.5 | 12.3 | 14.2 | 12.4 | 7.5 | 8.3 |
| $E$ | 6.5 | 7.2 | 8.9 | 7.4 | 4.5 | 5.9 |
| $F$ | 0.8 | 1.4 | 2.8 | 1.2 | 0.2 | 0.5 |
| *Majority Word Baseline* | | | | | | |
| Major-Word | 0.3 | 0.2 | 0.3 | 0.4 | 0.3 | 0.3 |

**Spoken Word Classification**   Table 3 presents our results on spoken word classification. We observe that the accuracy decreases as the level of supervision decreases, as expected. We also note that although the Word Classifier significantly outperforms all the other approaches under all corpora settings, the prerequisite for training such a fully-supervised approach is unrealistic—it requires the utterances to be perfectly segmented into audio segments corresponding to words with the word identity of each segment *known*. We emphasize that the Word Classifier is just used to establish an upper bound performance that gives us an idea on how good the classification results could be.

For alignment-based approaches, configuration $A^*$ achieves the highest accuracies under all corpora settings by using a parallel dictionary as cross-modal supervision for learning the alignment. However, we see that configuration $A$ using our unsupervised alignment approach only suffers a slight decrease in performance, which demonstrates that our unsupervised alignment approach is almost as effective as it supervised counterpart $A^*$. As we move towards unsupervised methods (k-means clustering) for grouping embeddings, in configuration $B$, a decrease in performance is observed.

The performance of using unsupervised segmentation algorithms is behind using exact word segments for training Speech2Vec, shown in configurations $C, D$, and $E$ versus $B$. We hypothesize that word segmentation is a critical step, since incorrectly separated words lack a logical embedding, which in turn hinders the clustering process. The importance of proper segmentation is evident in configuration $F$ as it performs the worst.

The aforementioned analysis applies to different corpora settings. We also observe that the performance of the embeddings learned from different corpora is inferior to the ones learned from the same corpus (refer to columns 1 and 3, versus 5 and 6, in Table 3). We think this is because the embedding spaces learned from the same corpora (e.g., both embeddings were learned from LibriSpeech) exhibit higher similarity than those learned from different corpora, making the alignment more accurate.

**Spoken Word Synonyms Retrieval**   Word classification does not display the full potential of our alignment approach. In Table 4 we show a list of retrieved results of example input audio segments. The words were ranked according to the cosine similarity between their embeddings and that of the

audio segment mapped from the speech embedding space. We observe that the list actually contain both synonyms and different lexical forms of the audio segment. This provides an explanation of why the performance of alignment-based approaches on word classification is poor: the top ranked word may not match the underlying word of the input audio segment, and would be considered incorrect for word classification, despite that the top ranked word has high chance of being semantically similar to the underlying word.

Table 4: Retrieved results of example audio segments that are considered incorrect in word classification. The match for each audio segment is marked in bold.

| Rank | Input audio segments | | | |
|------|-----------|--------|----------|----------|
| | beautiful | clever | destroy | suitcase |
| 1 | lovely | cunning | destroyed | bags |
| 2 | pretty | smart | **destroy** | suitcases |
| 3 | gorgeous | **clever** | annihilate | luggage |
| 4 | **beautiful** | crafty | destroying | briefcase |
| 5 | nice | wisely | destruct | **suitcase** |

We define word synonyms retrieval to also consider synonyms as valid results, as opposed to the word classification. The synonyms were derived using another language as a pivot. Using the cross-lingual dictionaries provided by [14], we looked up the acceptable word translations, and for each of those translations, we took the union of their translations back to the original language. For example, in English, each word has 3.3 synonyms (excluding itself) on average. Table 5 shows the results of word synonyms retrieval. We see that our approach performs better at retrieving synonyms than classifying words, an evidence that the system is learning the semantics rather than the identities of words. This showcases the strength of our semantics-focused approach.

Table 5: Results on spoken word synonyms retrieval. We measure how many times one of the synonyms of the input audio segment is retrieved, and report precision@$k$ for $k = 1, 5$.

| Corpora | $\mathrm{EN_{ls}} - \mathrm{en_{ls}}$ | | $\mathrm{FR_{ls}} - \mathrm{fr_{ls}}$ | | $\mathrm{EN_{swc}} - \mathrm{en_{swc}}$ | | $\mathrm{DE_{swc}} - \mathrm{de_{swc}}$ | | $\mathrm{EN_{ls}} - \mathrm{en_{swc}}$ | | $\mathrm{EN_{swc}} - \mathrm{en_{ls}}$ | |
|---------|------|------|------|------|------|------|------|------|------|------|------|------|
| Average P@k | P@1 | P@5 | P@1 | P@5 | P@1 | P@5 | P@1 | P@5 | P@1 | P@5 | P@1 | P@5 |
| *Alignment-based approach with cross-modal supervision (parallel dictionary)* | | | | | | | | | | | | |
| $A^*$ | 52.6 | 66.9 | 46.6 | 69.4 | 47.4 | 62.5 | 49.2 | 63.7 | 41.3 | 54.2 | 39.0 | 49.4 |
| *Alignment-based approaches without cross-modal supervision (our approach)* | | | | | | | | | | | | |
| $A$ | 43.2 | 57.0 | 42.4 | 58.0 | 36.3 | 50.4 | 32.6 | 48.8 | 33.9 | 47.5 | 33.4 | 45.7 |
| $B$ | 35.0 | 48.2 | 35.4 | 50.4 | 33.8 | 44.6 | 29.3 | 45.4 | 30.0 | 42.9 | 31.1 | 40.7 |
| $C$ | 27.7 | 37.3 | 26.4 | 35.7 | 21.1 | 30.3 | 26.2 | 34.5 | 22.4 | 28.9 | 17.1 | 26.3 |
| $D$ | 26.7 | 35.2 | 27.2 | 36.3 | 21.1 | 28.2 | 25.3 | 33.2 | 21.2 | 29.3 | 18.7 | 25.1 |
| $E$ | 17.7 | 24.2 | 20.8 | 28.4 | 17.3 | 21.8 | 18.3 | 23.0 | 15.2 | 21.1 | 11.2 | 17.8 |
| $F$ | 3.5 | 5.7 | 5.2 | 6.9 | 3.8 | 5.8 | 2.7 | 4.9 | 3.2 | 5.7 | 2.9 | 4.4 |

**Spoken word translation**  Table 6 presents the results on spoken word translation. Similar to spoken word classification, configurations with more supervision yield better performance than those with less supervision. Furthermore, we observe that translating using the same corpus outperforms those using different corpora (refer to $\mathrm{EN_{swc}} - \mathrm{de_{swc}}$ versus $\mathrm{EN_{ls}} - \mathrm{de_{swc}}$). We attribute this to the higher structural similarity between the embedding spaces learned from the same corpora.

# 6   Conclusions

In this paper, we propose a framework capable of aligning speech and text embedding spaces in an unsupervised manner. The method learns the alignment from independent corpora of speech and text, without requiring any cross-modal supervision, which is especially important for low- or zero-resource languages that lack parallel data with both audio and text. We demonstrate the effectiveness of our unsupervised alignment by showing comparable results to its supervised alignment counterpart

Table 6: Results on spoken word translation. We measure how many times one of the correct translations of the input audio segment is retrieved, and report precision@$k$ for $k = 1, 5$.

| Corpora | $\mathrm{EN_{ls}} - \mathrm{fr_{ls}}$ | | $\mathrm{FR_{ls}} - \mathrm{en_{ls}}$ | | $\mathrm{EN_{swc}} - \mathrm{de_{swc}}$ | | $\mathrm{DE_{swc}} - \mathrm{en_{swc}}$ | | $\mathrm{EN_{ls}} - \mathrm{de_{swc}}$ | | $\mathrm{FR_{ls}} - \mathrm{de_{swc}}$ | |
|---|---|---|---|---|---|---|---|---|---|---|---|---|
| Average P@k | P@1 | P@5 | P@1 | P@5 | P@1 | P@5 | P@1 | P@5 | P@1 | P@5 | P@1 | P@5 |
| *Alignment-based approach with cross-modal supervision (parallel dictionary)* | | | | | | | | | | | | |
| A* | 47.9 | 56.4 | 49.1 | 60.1 | 40.2 | 51.9 | 43.3 | 55.8 | 34.9 | 46.3 | 33.8 | 44.9 |
| *Alignment-based approaches without cross-modal supervision (our approach)* | | | | | | | | | | | | |
| A | 40.5 | 50.3 | 39.9 | 50.9 | 32.8 | 43.8 | 33.1 | 43.4 | 31.9 | 42.2 | 30.1 | 42.1 |
| B | 36.0 | 44.9 | 35.5 | 44.5 | 27.9 | 38.3 | 30.9 | 40.9 | 26.6 | 35.3 | 25.4 | 38.2 |
| C | 24.7 | 35.4 | 23.9 | 37.3 | 22.0 | 30.3 | 20.5 | 29.1 | 19.2 | 26.1 | 14.8 | 23.1 |
| D | 25.4 | 33.1 | 24.4 | 34.6 | 23.5 | 29.1 | 20.7 | 31.3 | 20.8 | 25.9 | 14.5 | 22.4 |
| E | 15.4 | 20.6 | 16.7 | 19.9 | 14.1 | 15.9 | 16.6 | 17.0 | 14.8 | 16.7 | 9.7 | 11.8 |
| F | 4.3 | 5.6 | 6.9 | 7.5 | 4.9 | 6.5 | 5.3 | 6.6 | 4.2 | 5.9 | 1.8 | 2.6 |
| *Majority Word Baseline* | | | | | | | | | | | | |
| Major-Word | 1.1 | 1.5 | 1.6 | 2.2 | 1.2 | 1.5 | 2.0 | 2.7 | 1.1 | 1.5 | 1.6 | 2.2 |

that uses full cross-modal supervision ($A$ vs. $A^*$) on the tasks of spoken word classification and translation. Future work includes devising unsupervised speech segmentation approaches that produce more accurate word segments, an essential step to obtain high quality speech embeddings. We also plan to extend current spoken word classification and translation systems to perform standard ASR and speech-to-text translation, respectively.

**Acknowledgments**

The authors thank Hao Tang, Mandy Korpusik, and the MIT Spoken Language Systems Group for their helpful feedback and discussions.

## Footnotes

[1]We tried window size $k \in \{1, 2, 3, 4, 5\}$ and embedding dimension $d \in \{50, 100, 200, 300\}$ and found that the reported $k$ and $d$ yield the best performance

[2]We also tried multi-layer neural network to model $W$. However, we did not observe any improvement on our evaluation tasks when using it compared to a linear $W$. This discovery aligns with [5].

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
