[Reviews · NeurIPS 2018]

Reviewer 1



This paper investigates unsupervised alignment between text and speech using embedding for both modalities. Word and speech embedding are first generated, the speech embedding are computed using an unsupervised alignment procedure and averaged. The alignment model is then trained based on adversarial training. The proposed approach is then evaluated on two tasks: spoken word detection and spoken word translation. Quality: The results of the spoken word detection study are not very appealing: 89% accuracy for the supervised baseline versus 23.7% for the unsupervised proposed approach is rather poor. But the synonym retrieval results are very interesting. Showing that the speech embedding seems to capture semantics information instead of phonetic is significant. The spoken word translation experiment is also interesting. For the spoken word detection task, it's not very clear: what is the input exactly ? is it a full utterance containing several words including the target one, or was the utterance aligned and cropped to contain only the uttered word ? I also think the authors should select a more established task instead of the one they proposed, spoken term detection seems relevant [1]. Clarity: the paper is well-written and easy to read. Originality: This work is a novel combination of established techniques, such as word embedding and adversarial training. The unsupervised method to compute speech embedding is novel to my knowledge. Significance: Unsupervised sequence classification, such as ASR and MT, is definitely the next huge milestone in the field, and this paper is significant step towards that long-term objective. Overall, despite the not-so-great results, I think the paper should be accepted as it presents a good lead toward unsupervised sequence classification. [1] J. G. Fiscus, J. Ajot, J. Garofolo, and G. Doddingtion, “Results of the 2006 spoken term detection evaluation,” in Proc. SIGIR, vol. 7. Citeseer, 2007, pp. 51–57. UPDATE: The A vs A* result is very interesting and is a very good result to demonstrate the potential of the approach. I suggest that this result should be better highlighted in the final version paper

Reviewer 2



The authors build off of well established techniques in unsupervised alignment between word embeddings of different languages to do unsupervised alignment between speech embeddings and word embeddings. They demonstrate this allows for some level of automatic word recognition and speech to translated text, without any labels. They also find that the biggest factor impacting performance is an unsupervised word parsing of the raw audio to begin with (chunking). The experiments are performed well, and the ablation analysis on the different alignment techniques is very clear. The technique is not very original, as it's a logical combination of pre-existing ideas, but that's not a problem as long as it's shown that the results are significant. Unfortunately, it's unclear in the current text how to evaluate the promise of this approach, as only internal comparisons are provided. The unsupervised approach obviously underperforms the supervised classifier, but it would be much more helpful to have some other semisupervised ASR techniques to compare against. The "failure" by producing synonyms seems like a strength of this technique, but it's hard to say without a comparison to a more traditional semisupervised baseline. Given the lack of new techniques, these additional experiments could be helpful for the authors to make their case as to the potential of this approach to semi/un supervised speech-text conversion. The authors also use k-means to cluster the audio embeddings, but the need to do so is unclear as they are used in a continuous value regression problem, and the task is to select the nearest text token, not audio token, so I'm not sure why there has to be an audio dictionary. It's seems like an unnecessary compression of the space that the authors find to hurt the performance.

Reviewer 3



The paper is about unsupervised alignment of speech and text embedding spaces. The paper basically connects 2 papers: (i) Speech2Vec, (ii) ICLR 2018 paper on unsupervised alignment of word embedding spaces of two languages. The idea is that a cross-modal alignment can allow to carry out retrieval across the speech and text modalities which can be useful for speech recognition (in one or other languages). The experimental setup is very well executed and goes from varying level of supervision used to get speech embeddings (dependent on the segmentation algorithms used). The authors also carry out the synonym retrieval analysis pointing to, as expected, semantic nature of the embeddings as also observed in (Kamper, Interspeech 2017). Overall, a really nicely executed paper! Two suggestions: (a) Can the authors include the majority word baseline i.e. for both spoken word classification & translation predict the most commonly paired word. (b) Some of the references ([14], [35]) are missing author names. ************** After rebuttal ***************** Thanks for answering all queries by me and my fellow reviewers. One suggestion for future work that I have after a second read is how do you minimize the gap between A* and supervised methods and hopefully later a trickle down effect to unsupervised approaches. As footnote 2 says, a more complex transformation between the embedding spaces doesn't really help. I'm pretty sure the problem is with L2 loss function where you're penalizing for not exactly matching the text embedding, but what you really care about is ranking. For the supervised approach(es), it's easier to try other losses (triplet etc) but not really sure about how to do it for unsupervised approaches. Might be a good avenue to work in future work.